# Association of Obesity and Malnutrition with In-Hospital Mortality and Clinical Outcomes in Patients Receiving Maintenance Dialysis: A National Database Study

**DOI:** 10.3390/nu18010157

**Published:** 2026-01-02

**Authors:** Wannasit Wathanavasin, Wisit Kaewput, Charat Thongprayoon, Supawit Tangpanithandee, Supawadee Suppadungsuk, Wisit Cheungpasitporn

**Affiliations:** 1Division of Nephrology and Hypertension, Department of Medicine, Mayo Clinic, Rochester, MN 55905, USA; wannasit.medical@mfu.ac.th (W.W.); supawit_d@hotmail.com (S.T.); s.suppadungsuk@hotmail.com (S.S.); cheungpasitporn.wisit@mayo.edu (W.C.); 2Nephrology Unit, Department of Medicine, Charoenkrung Pracharak Hospital, Bangkok Metropolitan Administration, Bangkok 10120, Thailand; 3Department of Military and Community Medicine, Phramongkutklao College of Medicine, Bangkok 10400, Thailand; wisitnephro@gmail.com; 4Chakri Naruebodindra Medical Institute, Faculty of Medicine Ramathibodi Hospital, Mahidol University, Samut Prakan 10540, Thailand

**Keywords:** end-stage kidney disease, in-hospital outcomes, malnutrition, nationwide study, obesity paradox

## Abstract

**Background/Objectives**: The study aimed to investigate whether malnutrition influences the obesity paradox and to explore the relationship between obesity with/without malnutrition and in-hospital outcomes among hospitalized ESKD patients. **Methods**: The study used the National Inpatient Sample database from 2016 to 2021. Hospitalized ESKD patients were included and categorized into three groups: non-obese, obese without malnutrition, and obese with malnutrition. The association between obesity with/without malnutrition and in-hospital outcomes, compared to non-obese patients, were analyzed. **Results**: Of 674,367 hospitalized ESKD patients included, 125,978 (18.7%) had obesity. Obese ESKD patients without malnutrition were associated with a decreased risk of mortality (odd ratio [OR] 0.87, 95% CI 0.84–0.91), whereas obese patients with malnutrition were associated with an increased risk of mortality (OR 2.08, 95% CI 1.90–2.27), compared to non-obese patients. Furthermore, obesity, with or without malnutrition, was linked to higher infection-related complications and resource utilization, especially when malnutrition was present. **Conclusions**: Our findings show that obesity is significantly associated with lower in-hospital mortality among ESKD patients without malnutrition. However, when malnutrition coexists, this survival advantage is reversed, underscoring the importance of detecting malnutrition in obese ESKD patients.

## 1. Introduction

End-stage kidney disease (ESKD) remains a challenge to global healthcare systems. Among the 850 million individuals affected by chronic kidney disease (CKD) globally, over 2.5 million have ESKD, with most of them requiring maintenance dialysis therapy [1]. These patients are at an extremely high risk of hospitalization [2] and mortality [3,4]. Of the various risk factors for disease complications, nutritional imbalance is considered one of the most critical issues. Malnutrition is prevalent and served as a significant predictor of mortality in ESKD patients [5,6,7]. Several potential pathophysiologic mechanisms underlie this, which could be classified into two main categories: those resulting from the disease itself, such as fluid and electrolyte imbalances that impact appetite and nutrient absorption, and uremic-related gastrointestinal symptoms that affect dietary intake [8]; and those resulting from dialysis therapy, including increased protein loss in the dialysate [9] and higher energy expenditure [10]. Furthermore, hospitalization for any illness in this population further heightens their susceptibility to malnutrition [11], in addition to the mechanisms previously mentioned.

Malnutrition is typically regarded as a condition of undernutrition, leading to the assumption that it only impacts underweight patients. However, the World Health Organization (WHO) states that malnutrition is also found in overweight and obese individuals [12], with a significant proportion, due to poor micronutrient intake and food quality. It is widely accepted that obesity is associated with a higher risk of various diseases which include, but not are limited to, diabetes mellitus and hypertension [13], as well as an increased risk of mortality [14]. However, in those who have experienced chronic disease, particularly ESKD, it seems to correlate with better survival outcomes, a phenomenon referred to as the “obesity paradox” [15,16]. Recent evidence suggest that both higher muscle mass and increased total body fat may provide protective effects [15]. Other supported hypotheses include obesity being associated with better short-term hemodynamic stability, higher levels of cardio-protective markers like soluble tumor necrosis factor-alpha (TNF-alpha) receptors, and increased sequestration of uremic toxins in adipose tissue [17], as shown in Figure 1.

When considering the coexistence of these two nutritional disorders (obesity and malnutrition) within one individual at the same time, there is limited evidence regarding their impact on in-hospital clinical outcomes and resource utilization among ESKD patients. To bridge this knowledge gap, we aim (1) to investigate whether malnutrition influences the obesity paradox (survival benefits), and (2) to explore the relationship between obesity with/ without malnutrition and in-hospital outcomes among hospitalized ESKD patients on maintenance dialysis.

## 2. Materials and Methods

### 2.1. Dataset

The National Inpatient Sample (NIS) was used in this study. The NIS supplies data on US regional and national estimates of inpatient utilization, access, charges, quality, and outcomes. It is derived from all states participating in Healthcare Cost and Utilization Project (HCUP), encompassing over 97% of the US population. The NIS ensures patient confidentiality by omitting state and hospital identifiers from the publicly accessible dataset. Institutional review board approval was waived, as the database includes no patient-identifiable data.

### 2.2. Study Population

We used NIS data from 2016 to 2021 and identified adult patients with ESKD patients on maintenance dialysis. To determine patient eligibility and categorized patients, the International Classification of Diseases, Tenth Revision, Clinical Modification (ICD-10-CM) code were employed. We included all hospitalized ESKD patients receiving HD and PD. Patients with kidney transplants, acute kidney injury, or non-dialysis dependent CKD (NND-CKD) were excluded. Obesity and malnutrition were identified based on ICD-10 codes, with malnutrition classifications including all degrees of severity, from mild to severe. Appendix A showed information on the ICD-10 codes used in this analysis.

### 2.3. Study Variables and Outcomes

Patients’ baseline characteristics included sex, age, race, mode of kidney replacement therapy (KRT), comorbidity measures, comorbidities, and history of smoking and alcohol consumption. In addition, the type of admission, hospital type, and teaching status were extracted from the NIS database.

The study outcomes were the associations between obesity, with or without malnutrition, and in-hospital outcomes among patients with ESKD, compared with non-obese patients. These outcomes included in-hospital mortality, adverse clinical events, inpatient treatments, and resource utilization, including hospitalization costs and length of stay.

### 2.4. Statistical Analyses

Covariates and outcomes were compared among the obesity without malnutrition, obesity with malnutrition, and non-obese groups using one-way ANOVA for normally distributed continuous variables, the Kruskal–Wallis test for skewed continuous variables, and the Chi-squared test for categorical variables. Categorical variables are summarized as unweighted counts (n) and percentages (%). Normally distributed continuous variables are presented as means with standard deviations (SD), whereas non-normally distributed continuous variables are reported as medians with interquartile ranges (IQR). Logistic regression analyses were used to evaluate the associations between obesity with or without malnutrition and adverse clinical outcomes, as well as inpatient treatments. Linear regression models were applied to assess the relationship between obesity with or without malnutrition and measures of resource utilization. All models were adjusted for potential patient-level confounders, including age, sex, race, KRT modality, smoking and alcohol use, admission type, comorbidities, and hospital-level characteristics. We conducted a subgroup analysis stratified by dialysis modality and tested for interactions between dialysis modality (HD vs. PD) and obesity/malnutrition category.

## 3. Results

### 3.1. Baseline Characteristics

Of 674,367 hospitalized ESKD patients receiving dialysis, 125,978 (18.7%) had obesity diagnosis. Among those patients with obesity, 119,155 (94.6%) were not malnourished, while 6823 (5.4%) had malnutrition. Table 1 details the demographic characteristics across obese ESKD patients with malnutrition, without malnutrition, and non-obese patients. Overall, significant differences in baseline patient characteristics were observed among the three groups (*p* < 0.05). Compared to non-malnourished obese ESKD patients, those with malnourished obesity were more likely to be older, female, White, receiving PD modality, and have a history of alcohol consumption. Additionally, they had a higher prevalence of chronic conditions, including peripheral vascular disease, cerebrovascular disease, cirrhosis, dementia/cognitive impairment, and cancer, resulting in a higher Elixhauser Comorbidity index.

### 3.2. Association Between Obesity with/Without Malnutrition and In-Hospital Outcomes

#### 3.2.1. In-Hospital Mortality

The in-hospital mortality among ESKD patients from 2016 to 2021 was 4.6%. Mortality was higher in the malnourished obesity group (9.8%) but lower in the non-malnourished obesity group (3.6%), compared with the non-obesity group (4.7%) (*p* < 0.001). In adjusted analyses, malnourished obesity was independently associated with an increased risk of in-hospital death (OR 2.08), whereas non-malnourished obesity was linked to a decreased risk (OR 0.87) compared to non-obese patients (Table 2).

#### 3.2.2. Adverse Clinical Outcomes

Among hospitalized ESKD patients, malnourished obesity was linked to higher risks of sepsis (OR 2.63) and catheter-related bloodstream infection (OR 1.70), but a lower risk of volume overload (OR 0.91). In contrast, non-malnourished obesity was associated with elevated risks for all adverse clinical outcomes, including sepsis (OR 1.07), catheter-related bloodstream infection (OR 1.09), and volume overload (OR 1.08), as shown in Table 2.

#### 3.2.3. Inpatient Treatments

In hospitalized ESKD patients, malnourished obesity was associated with greater risks of vasopressor administration (OR 2.56), TPN (OR 4.36), mechanical ventilation (OR 1.95), and blood transfusions (OR 1.60). Conversely, non-malnourished obesity was linked to lower risks of total parenteral nutrition (OR 0.57) and blood transfusions (OR 0.89), but higher risks of vasopressor use (OR 1.06) and mechanical ventilation (OR 1.85) versus non-obese patients (Table 2).

#### 3.2.4. Resource Utilization

ESKD patients with malnourished obesity significantly had a longer length of hospital stay by 7.14 days and higher hospitalization cost by $99,514 compared to ESKD patients without obesity. Similarly, patients with non-malnourished obesity, though to a lesser extent, had a longer length of hospital stay by 0.14 days and higher hospitalization costs by $2811 than patients without obesity (Table 2).

#### 3.2.5. Subgroup Analysis Stratified by Dialysis Modality

Appendix A summarizes the association between obesity, with or without malnutrition, and in-hospital outcomes stratified by dialysis modality. In brief, multivariable analysis showed no significant interaction between malnutrition/obesity status and dialysis modalities on in-hospital outcomes, including in-hospital mortality, sepsis, catheter-related bloodstream infection, and the need for vasopressors, total parenteral nutrition, or blood transfusions (all *p* for interaction > 0.05; Appendix A). However, malnourished obese patients were associated with a lower risk of volume overload in HD patients (OR 0.89) but not in PD patients (OR 1.37). Additionally, significant differences were observed between dialysis modalities with respect to mechanical ventilation requirements and resource utilization. Specifically, patients receiving HD showed a more pronounced higher need for mechanical ventilation, longer hospital stays, and higher hospitalization costs than those receiving PD (all *p* for interaction < 0.05; Appendix A).

## 4. Discussion

This large-scale, nationwide US study of over half a million ESKD patients observes the protective association of obesity on in-hospital mortality outcome among ESKD patients without malnutrition, illustrating the “obesity paradox”. However, when malnutrition coexists with obesity, this protective association is reversed, resulting in conversely increased mortality. Furthermore, after adjusting for relevant confounders, our study demonstrated that obesity, with or without malnutrition, was linked to a higher risk of catheter-related bloodstream infection and sepsis, a greater need for inpatient treatments with vasopressors and mechanical ventilators, and higher hospital length of stay and hospital costs. These effects were notably more pronounced when malnutrition was present. In contrast to obese ESKD patients with malnutrition, those without malnutrition require less use of TPN and blood transfusions. In terms of volume overload outcomes, the presence of malnutrition in obese ESKD, particularly those receiving HD, is linked to a reduced risk, whereas the risk increases for those without malnutrition (Table 2).

Despite obesity and malnutrition being common among hospitalized patients, they are rarely considered together. As a result, recent evidence on the coexistence of these two nutritional disorders and their association with clinical outcomes, especially in-hospital mortality and adverse in-hospital outcomes in ESKD patients, is limited. One of the reasons is that malnutrition in patients with low or normal BMI is easily identified, but in obese patients, it is often overlooked as their fat mass masks the deterioration of muscle mass. Our results highlight that obese ESKD patients may also suffer from malnutrition, putting them at a higher mortality risk linked to malnutrition, diminishing the obesity paradox effect (Table 2). Similarly, Robinson et al. [18] suggest that malnutrition should be considered as a critical determinant in the association between obesity and mortality. When nutritional variables are not included in the statistical analyses, obesity appears to be protective for critically ill patients in the intensive care unit (ICU). However, after adjusting for nutritional status and nutritional markers, a higher mortality risk is observed in obese patients (OR 1.58; 95% CI 1.21–2.07 for malnutrition and OR 2.67; 95% CI 2.06–3.44 for serum albumin < 3.4 g/dL). These findings imply that conventional diagnostic methods, using phenotypic criteria such as low body mass index (BMI) and percentage weight loss, may lead to a malnutrition under-diagnosis. Moreover, BMI measurement is insufficient to identify changes in body composition, especially reduced skeletal muscle mass and function (sarcopenia or sarcopenic obesity), which are crucial nutritional indicators in predicting mortality risk among patients undergoing dialysis [19,20]. Obesity is a highly heterogeneous condition and can be classified into distinct phenotypes based on fat distribution [21]. BMI is also limited in its ability to characterize regional adiposity. Visceral fat has been shown to be a more clinically relevant predictor of prognosis than subcutaneous fat. According to Iida et al. [22], chronic HD patients, particularly men, with a high visceral fat area-to-abdominal fat ratio have a significantly increased risk of mortality. Existing nutritional screening approaches appear insufficient to comprehensively evaluate the complex nutritional challenges experienced by obese patients with ESKD [23]. Further research is needed to identify optimal assessment tools or nutritional biomarkers with clinical applicability for early detecting malnutrition in this complex population.

Although we adjusted for multiple potential confounders, several clinically relevant factors influencing the association between malnutrition and mortality in dialysis patients were not captured in this nationwide database, including residual kidney function (RKF), dialysis modality, severity of inflammation, and nutritional interventions. RKF is a strong predictor of survival, contributing to fluid and electrolyte balance and the clearance of middle molecules and protein-bound uremic toxins, with retention of these toxins increasing cardiovascular and infection-related mortality [24]. Patients with preserved RKF generally have better appetite and a less restricted diet, which may reduce the risk of malnutrition, whereas those without RKF often exhibit higher inflammatory markers [25], exacerbating the malnutrition–inflammation complex. Furthermore, dialysis modality (such as conventional HD or hemodiafiltration) and standard indicators of dialysis adequacy, including Kt/V, urea reduction ratio (URR), and delivered dose, could not be determined in this dataset. Recent evidence suggests hemodiafiltration may improve survival compared with conventional high-flux HD [26]; therefore, unmeasured dialysis modality and adequacy of dialysis—particularly in patients with obesity—remains a potential source of residual confounding. Moreover, chronic inflammation in the dialysis population may play a central role in modifying the obesity paradox effect. Persistent inflammation can promote muscle wasting, hypoalbuminemia, and anorexia, as well as impair cardiac contractility and accelerate atherosclerotic vascular disease, a constellation commonly referred to as the malnutrition–inflammation–atherosclerosis (MIA) syndrome [27], which is associated with increased mortality risk. Because direct measures of inflammatory markers were not available in this dataset, ICD-10-based diagnoses of malnutrition may, in part, reflect inflammation-driven catabolism rather than nutritional status alone. Lastly, nutritional interventions, such as oral nutritional supplementation, can mitigate malnutrition and improve survival outcomes, particularly in hypoalbuminemic HD patients [28]. Well-designed future studies incorporating these factors are warranted to better elucidate the relationship between malnutrition and mortality in dialysis patients.

The mechanisms underlying the development of malnutrition that co-exists with obesity in ESKD patients are complex and represent a multi-faceted interplay among poor diet quality, chronic low-grade inflammation, and the malabsorption of micronutrients caused by various kidney medications [29]. We hypothesize that dietary patterns dominated by ultra-processed foods (UPFs), which are energy-dense but low in fiber and essential micronutrients, could contribute to this dual burden. Given that recently UPFs provide over 50% of total caloric intake in the U.S. and parallel the rise in obesity rates [30], it is plausible that high UPF consumption may predispose CKD patients to uremic metabolic derangements or exacerbate existing nutritional deficits [31]. Conversely, less processed, fiber-rich diets might counteract these effects, potentially mitigating adverse clinical outcomes in this population [32,33,34]. Further investigation is warranted to test these potential mechanistic links and their impact on patient outcomes. Furthermore, our study observes that obese ESKD patients, regardless of nutritional status, experience a higher incidence of infectious complications, such as sepsis and CRBSI, and appear to face more severe critical conditions, requiring increased use of vasopressors and mechanical ventilation (Table 2). Likewise, a study by Dossett et al. examined the relationship between BMI and the risk of site-specific ICU-acquired infections in critically ill adults. They found that severe obesity is an independent risk factor for CRBSI, with an odds ratio of 3.20 (95%CI 1.90, 5.30) among surgical ICU patients. The vicious cycle of being more susceptible to infections worsens nutritional status and aggravates existing malnutrition in individuals by reducing nutrient intake, interfering with substrate utilization, and promoting tissue breakdown [35].

As expected, our results show that obesity accompanied by malnutrition is more strongly associated with worsening of most adverse clinical outcomes compared to obesity without malnutrition, except for volume overload. Unlike earlier research [36], our findings indicate that malnutrition actually mitigates the risk of volume overload, particularly those receiving HD (Table 2 and Appendix A). The exact reason for this finding is unclear, but it might be explained by differences in baseline characteristics, as the obesity with malnutrition group had lower rates of coronary artery disease when compared to other groups (Table 2). These characteristics typically reduce susceptibility to volume overload. Though we attempted to adjust for these potential confounders through multivariable analysis, the protective association persisted with borderline significance. This finding suggests the presence of possible unmeasured confounders that could not be entirely accounted for given our data limitations. These factors include RKF, dialysis modality, and prescription parameters, along with nutritional interventions, all potentially affecting patients’ fluid balance. However, it is important to recognize that volume overload in dialysis population stems from multiple complex causes. Additional studies that incorporate and adjust for all of these potential confounding factors are necessary to better understand these intricate relationships.

Among hospitalized ESKD patients, the presence of obesity coexisting with malnutrition was associated with a significantly higher demand for in-hospital treatments and resource utilization compared to patients without obesity or those with obesity alone, without malnutrition (Table 2). This population required nearly five times more nutritional support, particularly TPN use, as demonstrated by our study. According to The European Society for Clinical Nutrition and Metabolism (ESPEN) guideline [37], hospitalized ESKD patients who are unable to achieve at least 70% of their macronutrient requirements through oral nutrition should receive parenteral nutrition (PN), regardless of whether they are critically or non-critically ill. As mentioned earlier, obese ESKD patients with malnutrition tend to experience more severe conditions, which may interfere with their ability to maintain oral intake, particular in cases of unstable hemodynamics or bowel ileus from sepsis, thereby increasing the requirement for TPN. Blood transfusion is another inpatient treatment that should be highlighted, as it is more common in obese individuals with malnutrition (Table 2). The influence of malnutrition to anemia is more significant than expected, with deficiencies in iron, folate, and vitamin B12 causing nutritional anemia [38]. Kalantar-Zadeh et al. [39] emphasized that in ESKD patients, the malnutrition–inflammation complex syndrome (MICS) can further impair the response to recombinant human erythropoietin in treating renal anemia. Moreover, refractory anemia is more frequently observed in dialysis patients with MICS, which helps explain the higher demand for blood transfusions in malnourished ESKD patients.

In subgroup analyses, obese individuals with malnutrition receiving HD experienced higher resource utilization than those receiving PD, as reflected by longer hospital stays and higher total hospital costs (Appendix A). This may be partly explained by the significantly higher need for mechanical ventilation among HD patients (*p* for interaction < 0.05), along with trends toward higher rates of sepsis and greater use of vasopressors and total parenteral nutrition, despite non-significant interaction testing. Together, these findings indicate a higher severity of illness among HD patients, with a greater likelihood of inpatient care unit (ICU)-level care. Nevertheless, when examining hard clinical outcomes, no significant difference in mortality was observed between the two modalities (Appendix A). Consistent with our findings, a propensity score-based outpatient analysis by Elsayed et al. [40] reported comparable survival between patients treated with PD and those receiving in-center HD, implying that dialysis modality may not be a major determinant of mortality outcomes in either inpatient or outpatient settings.

Consistent with earlier research [41], our findings found that well-nourished obese patients required fewer blood transfusions compared to both non-obese patients and obese patients with malnutrition. This protective association of elevated BMI on transfusion risk may be attributed to these patients’ greater baseline estimated blood volume. Despite obese patients often experiencing more severe critical illness, which can worsen anemia of inflammation and stress-related gastrointestinal bleeding, they still demonstrate a comparatively lower relative need for transfusions than non-obese individuals. The negative impact of obesity on length of stay and hospital expenses was observed regardless of nutritional status (Table 2); however, this effect was markedly intensified when malnutrition was concurrently present. Taken together, our results dispute the widespread assumption that obesity equates to adequate nutrition. All hospitalized ESKD patients should receive comprehensive nutritional screening, regardless of their BMI classification.

To our knowledge, this is the first and largest nationally representative study to examine comprehensive in-hospital outcomes, treatments, and healthcare resource utilization among dialysis patients. Our large sample size significantly strengthens the study’s statistical reliability. In addition, we offer several practical applications. First, our findings demonstrate that malnutrition can exist alongside obesity, and this combination is associated with a reduction in obesity’s survival advantage and negatively affecting nearly all aspects of inpatient care and resource use. Healthcare providers should focus on early malnutrition identification with comprehensive nutritional assessment rather than relying on only weight change or BMI classification. Second, our results emphasize the necessity for specialized nutritional screening protocols for obese ESKD patients to enable timely nutritional interventions and reduce disease burden. Finally, public health strategies should focus on preventing malnutrition in all ESKD patients, including those with obesity, to address the root cause of these complications.

The results of our study are not without limitations. First, its observational design limits the ability to establish causal relationships. Although extensive adjustments were made for potential confounders, residual confounding remains a possibility. Second, the lack of identifiable patient information limits our ability to assess the utilization of out-of-hospital utilization of other healthcare services, including ambulatory clinic or emergency department. As a result, we cannot evaluate out-of-hospital mortality rate and post-discharge death attributable to ESKD and obesity comorbidities. Moreover, readmission events cannot be identified; therefore, individual patients who experience multiple hospitalizations within a single year may be represented more than once in the dataset. Finally, the study relies on clinical coding standards, which follow ICD-10-CM diagnostic and procedural coding, potential undercoding or coding errors are unavoidable. Furthermore, the diagnostic approach to malnutrition could not be clearly delineated due to the absence of reviewable clinical data in the NIS database, including laboratory findings, nutritional assessment tools, established criteria, or other relevant clinical information. Differences and subjectivity in how hospitals apply malnutrition diagnostic criteria may allow financial incentives to influence diagnosis. Without clear standardization, malnutrition diagnoses could rise without corresponding clinical relevance and be more frequently assigned by hospitals with greater investment in coding and billing, increasing the likelihood of false-positive classification. This limitation may potentially introduce bias.

## 5. Conclusions

Our findings show that obesity is significantly associated with lower in-hospital mortality among ESKD patients without malnutrition. The presence of malnutrition, however, is associated with higher mortality, more severe infection-related complications, and increased resource utilization (Figure 2), highlighting the importance of identifying malnutrition in obese ESKD patients. Future research is needed to establish validated diagnostic criteria for malnutrition in this specific high-risk population.

## Figures and Tables

**Figure 1 nutrients-18-00157-f001:**
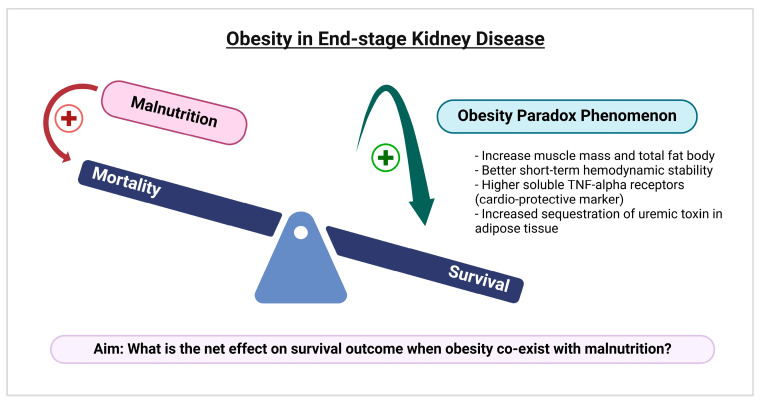
A conceptual framework illustrating the potential mechanisms behind the obesity paradox in end-stage kidney disease and the aim of our study to examine how the coexistence of malnutrition and obesity affects survival outcomes. This picture was created with biorender.com.

**Figure 2 nutrients-18-00157-f002:**
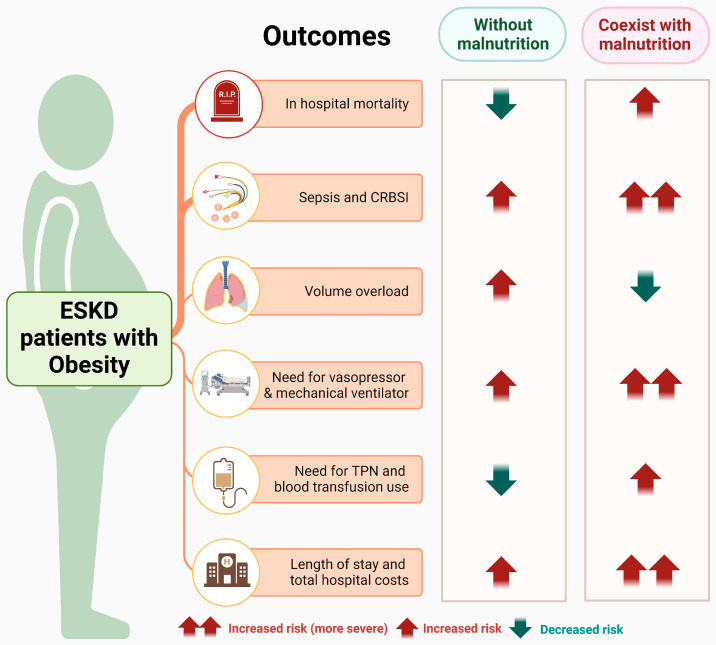
Summary of the association between obesity with/without malnutrition and in-hospital outcomes among hospitalized ESKD patients. Abbreviations: ESKD, end-stage kidney disease; TPN, total parenteral nutrition. This figure was generated with biorender.com.

**Table 1 nutrients-18-00157-t001:** Baseline characteristics of ESKD patients by malnutrition and obesity status.

Variables	Non-Obese (n = 548,389)	Obesity		*p*-Value
		Without Malnutrition (n = 119,155)	With Malnutrition (n = 6823)	
Age (years)	62.1 ± 15.5	59.9 ± 13.1	62.5 ± 12.8	<0.001
Male sex, n (%)	309,052 (56.4)	56,490 (47.4)	2920 (42.8)	<0.001
Race, n (%) - White - Black - Hispanic - Asian or Pacific Islander	209,489 (40.7) 184,923 (35.9) 94,710 (18.4) 25,404 (4.9)	51,537 (45.8) 40,123 (35.6) 18,085 (16.1) 2877 (2.6)	3130 (48.9) 2120 (33.1) 993 (15.5) 162 (2.5)	<0.001
Mode of KRT, n (%) - Hemodialysis - Peritoneal dialysis	508,983 (92.8) 39,406 (7.2)	111,288 (93.4) 7867 (6.6)	6267 (91.8) 556 (8.2)	<0.001
Elixhauser score, median (IQR)	6 (4–7)	7 (6–8)	8(7–10)	<0.001
Charlson comorbidity score, median (IQR)	5 (4–7)	6 (4–7)	6 (5–7)	<0.001
Comorbidity, n (%) - Diabetes mellitus - Hypertension - Dyslipidemia - Congestive heart failure - Coronary artery disease - Cerebrovascular disease - Peripheral vascular disease - Cancer - Cirrhosis - Dementia/cognitive impairment	338,002 (61.6) 521,290 (95.1) 218,487 (39.8) 276,133 (50.3) 91,643 (16.7) 51,505 (9.4) 71,184 (13.0) 29,515 (5.4) 54,236 (9.9) 35,401 (6.5)	93,201 (78.2) 114,574 (96.2) 58,674 (49.2) 65,840 (55.3) 19,795 (16.6) 8895 (7.5) 13,247 (11.1) 4280 (3.6) 8748 (7.3) 3680 (3.1)	5225 (76.6) 6388 (93.6) 2970 (43.5) 3802 (55.7) 997 (14.6) 648 (9.5) 848 (12.4) 446 (6.5) 993 (14.6) 399 (5.9)	<0.001 <0.001 <0.001 <0.001 <0.001 <0.001 <0.001 <0.001 <0.001 <0.001
Smoking, n (%)	122,164 (22.3)	26,622 (22.3)	1083 (15.9)	<0.001
Alcohol use, n (%)	15,125 (2.8)	1944 (1.6)	217 (3.2)	<0.001
Elective admission type, n (%)	37,819 (6.9)	9956 (8.4)	485 (7.1)	<0.001
Length of stay, days median (IQR)	5 (3–9)	5 (3–9)	10 (5–19)	<0.001
Hospitalization cost ($) median (IQR)	54,157 (29,140–107,001)	58,611 (31,598–113,183)	109,981 (54,656–229,221)	<0.001
Hospital location/ teaching status, n (%) - Rural - Urban-nonteaching - Urban-teaching	25,195 (4.6) 99,559 (18.2) 423,635 (77.2)	6028 (5.1) 21,557 (18.1) 91,570 (76.8)	277 (4.1) 1341 (19.6) 5205 (76.3)	<0.001

**Table 2 nutrients-18-00157-t002:** The association between obesity with/without malnutrition (vs non-obese) and in-hospital outcomes in ESKD patients receiving maintenance dialysis.

In-Hospital Outcomes	Non-Obese (n = 548,389)	Obesity Without Malnutrition (n = 119,155)				Obesity with Malnutrition (n = 6823)			
		Univariable Analysis		Multivariable Analysis		Univariable Analysis		Multivariable Analysis	
		OR (95% CI)	*p*-Value	Adjusted OR * (95% CI)	*p*-Value	OR (95% CI)	*p*-Value	Adjusted OR * (95% CI)	*p*-Value
**Adverse clinical outcomes**									
In hospital mortality	Ref.	0.75 (0.73–0.78)	<0.001	0.87 (0.84–0.91)	<0.001	2.20 (2.02–2.39)	<0.001	2.08 (1.90–2.27)	<0.001
Sepsis	Ref.	1.01 (0.99–1.03)	0.12	1.07 (1.05–1.08)	<0.001	2.77 (2.63–2.91)	<0.001	2.63 (2.50–2.77)	<0.001
CRBSI	Ref.	1.09 (1.03–1.15)	0.003	1.09 (1.03–1.15)	0.003	1.68 (1.43–1.98)	<0.001	1.70 (1.44–2.00)	<0.001
Volume overload	Ref.	1.08 (1.05–1.10)	<0.001	1.08 (1.06–1.11)	<0.001	0.84 (0.77–0.92)	<0.001	0.91 (0.83–0.99)	0.03
**Inpatient treatments**									
Need for vasopressors	Ref.	0.96 (0.91–1.01)	0.10	1.06 (1.01–1.12)	0.02	2.80 (2.49–3.15)	<0.001	2.56 (2.26–2.89)	<0.001
TPN use	Ref.	0.49 (0.43–0.57)	<0.001	0.57 (0.50–0.66)	<0.001	4.83 (4.04–5.77)	<0.001	4.36 (3.63–5.24)	<0.001
Mechanical ventilation	Ref.	1.84 (1.79–1.89)	<0.001	1.85 (1.80–1.90)	<0.001	1.98 (1.82–2.16)	<0.001	1.95 (1.79–2.13)	<0.001
Blood transfusion	Ref.	0.82 (0.80–0.84)	<0.001	0.89 (0.87–0.91)	<0.001	1.61 (1.50–1.72)	<0.001	1.60 (1.49–1.72)	<0.001
		Coefficient (95% CI)	*p*-value	Adjusted coefficient * (95% CI)	*p*-value	Coefficient (95% CI)	*p*-value	Adjusted coefficient * (95% CI)	*p*-value
**Resource utilization**									
LOS (days)	Ref.	0.00 (−0.07, 0.06)	0.93	0.14 (0.08, 0.20)	<0.001	7.57 (7.12, 8.02)	<0.001	7.14 (6.69, 7.58)	<0.001
Hospitalization cost ($)	Ref.	−948 (−2385, 489)	0.20	2811 (1561, 4061)	<0.001	104,245 (95,581, 112,908)	<0.001	99,514 (90,932, 108,096)	<0.001

Abbreviation: CRBSI, catheter-related bloodstream infection; LOS, length of stay; TPN, total parenteral nutrition. * Adjusted for age, gender, race, hospitalization year, Charlson comorbidity score, diabetes mellitus, congestive heart failure, hypertension, cerebrovascular disease, coronary artery disease, peripheral vascular disease, cancer, cirrhosis, cognitive impairment, alcohol consumption, smoking, KRT mode and admission type, hospital location/ teaching status.

## Data Availability

The data used in this study are available from the corresponding author upon request.

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
