# Peer review of "Association of Obesity and Malnutrition with In-Hospital Mortality and Clinical Outcomes in Patients Receiving Maintenance Dialysis: A National Database Study"

_nutrients, 2026, doi:10.3390/nu18010157_

Round 1

Reviewer 1 Report

Comments and Suggestions for Authors

nutrients-4011449-peer-review-v1

The Impact of Malnutrition on the Obesity Paradox among Patients with End-Stage Kidney Disease Requiring Maintenance Dialysis

This study utilized the US National Inpatient Sample database from 2016 to 2021. Patients were categorized based on ICD-10 codes. Hospitalized end-stage kidney disease (ESKD) patients were included and categorized into three groups: non-obese, obese without malnutrition, and obese with malnutrition. 674,367 hospitalized ESKD patients were included, 125,978 (18.7%) had obesity. Among those with obesity, 119,155 (94.6%) were not malnourished, while 6,823 (5.4%) had malnutrition. Obese ESKD patients without malnutrition had a decreased risk of mortality (odd ratio [OR] 0.87, 95% CI 0.84-0.91), whereas obese patients with malnutrition had an increased risk of mortality (OR 2.08, 95% CI 1.90-2.27), compared to non-obese patients.

Comments

The value of the study is the large population studied. The main problem seems the reliability of the obtained data. Diagnostic criteria related to ICD-10 codes are not precise. Therefore, classifying obese (BMI>30 ?) subjects to malnourished (on what criteria?) or not-malnourished seems prone to an important bias. The Authors also admit that “BMI measurement is insufficient to identify changes in body composition, especially reduced skeletal muscle mass and function…”.

The primary outcomes of the study were the association between obesity with or without malnutrition (compared to non-obese patients) and in-hospital outcomes. It is not clear whether the data concern the first hospitalization during the 2016 to 2021 period. Out-of-hospital outcomes are also not known.

Other comments

Some similar data has been published recently:

Kaewput W, Thongprayoon C, Suppadungsuk S, Tangpanithandee S, Wathanavasin W, Qureshi F, Cheungpasitporn W. Impact of obesity on in-hospital outcomes in peritoneal dialysis patients: insights from a nationwide analysis. Int Urol Nephrol. 2025 Aug;57(8):2595-2601. doi: 10.1007/s11255-025-04438-w. 

Table 1. State clearly that p-values concern differences between non-malnourished obese ESKD patients and malnourished obesity ESKD patients.

Paragraph 3.2 seems to concern different database:

“Out of the 1,244,415 hospitalized advanced CKD patients, 131,372 were diagnosed with malnutrition, resulting in an overall prevalence rate of 10.6%...”

Whole manuscript needs careful edition, e.g. reference:

  1. Kalantar-Zadeh K, Rhee CM, Chou J, Ahmadi SF, Park J, Chen JL, et al. The Obesity Paradox in Kidney Disease: How to 425 Reconcile it with Obesity Management. Kidney Int Rep. 2017;2(2):271-81.

is repeated as ref, nr 17.

Author Response

comment 1: The value of the study is the large population studied. The main problem seems the reliability of the obtained data. Diagnostic criteria related to ICD-10 codes are not precise. Therefore, classifying obese (BMI>30 ?) subjects to malnourished (on what criteria?) or not-malnourished seems prone to an important bias.

Author's reply: Thank you very much for your insightful comment. As you noted, obesity was classified using a BMI threshold of ≥30 kg/m², irrespective of etiology, and was identified based on ICD-10 codes (details provided in Supplementary Table S1). With respect to malnutrition, we acknowledge that the diagnostic approach to malnutrition could not be delineated because the NIS database does not contain reviewable clinical data, including laboratory findings, nutritional assessment tools, established diagnostic criteria, or other relevant clinical information. Accordingly, we have emphasized this limitation and have further revised the manuscript in the limitation part as follows:

“Furthermore, the diagnostic approach to malnutrition could not be clearly delineated due to the absence of reviewable clinical data in the NIS database, including laboratory findings, nutritional assessment tools, established criteria, or other relevant clinical information. This limitation may potentially introduce bias.”

comment 2: The Authors also admit that “BMI measurement is insufficient to identify changes in body composition, especially reduced skeletal muscle mass and function…”.

Author's reply: Thank you very much for pointing this out. We have included the sentence you mentioned to emphasize the limitation of using BMI alone to identify malnutrition. The coexistence of obesity and malnutrition, as identified by ICD-10 codes in our study, underscores the limitations of relying solely on BMI, which may result in underdiagnosis of malnutrition and an inadequate assessment of body composition. We therefore highlighted this issue as a practical consideration for real-world clinical practice.

comment 3: The primary outcomes of the study were the association between obesity with or without malnutrition (compared to non-obese patients) and in-hospital outcomes. It is not clear whether the data concern the first hospitalization during the 2016 to 2021 period. Out-of-hospital outcomes are also not known.

Author's reply: Thank you for bringing this to our attention. With respect to the NIS database, readmission events cannot be identified; therefore, individual patients with multiple hospitalizations within a single year may be represented more than once in the dataset, as acknowledged in the official NIS documentation. We have incorporated this limitation into the manuscript’s limitations section. In addition, outcomes occurring outside the hospital setting cannot be captured by this database, as acknowledged in the limitations section.

“The lack of identifiable patient information limits our ability to assess the utilization of out-of-hospital utilization of other healthcare services, including ambulatory clinic or emergency department. As a result, we cannot evaluate out-of-hospital mortality rate and post-discharge death attributable to ESKD and obesity comorbidities. Moreover, readmis-sion events cannot be identified; therefore, individual patients who experience multiple hospitalizations within a single year may be represented more than once in the dataset.”

comment 4: Some similar data has been published recently:

Kaewput W, Thongprayoon C, Suppadungsuk S, Tangpanithandee S, Wathanavasin W, Qureshi F, Cheungpasitporn W. Impact of obesity on in-hospital outcomes in peritoneal dialysis patients: insights from a nationwide analysis. Int Urol Nephrol. 2025 Aug;57(8):2595-2601. doi: 10.1007/s11255-025-04438-w. 

Author's reply: Thank you very much for your comment. We acknowledge that our study and study by Kaewput et al. utilized data from the same source (the NIS database); however, the two studies differ substantially in terms of PICOS elements, study period, and overall conceptual framework, as outlined below.

Study by Kaewput et al.

Our study

Population

Patients receiving peritoneal dialysis only

Dialysis patients, including both hemodialysis and peritoneal dialysis

Intervention/ Exposure

Single exposure

- Obesity

Two exposure groups

- Obesity with malnutrition

- Obesity without malnutrition

Comparison

Non-obese

Non-obese

Outcomes

-Treatments; PD catheter adjustment, hemodialysis, mechanical ventilation

- Complications and outcomes; PD-related peritonitis, PD mechanical complication, Hyperkalemia, metabolic acidosis, volume overload, sepsis, ventricular arrhythmia/ cardiac arrest, in-hospital mortality

-Resource utilization; length of stay and hospitalization costs

- In hospital mortality

- Adverse clinical outcomes; sepsis, catheter-related bloodstream infections (CRBSI), and volume overload

- Inpatient treatments; use of vasopressors, total parenteral nutrition (TPN), mechanical ventilator and blood transfusions

-Resource utilization; length of stay and hospitalization costs

Study periods

2003-2018

2016-2021

comment 5: Table 1. State clearly that p-values concern differences between non-malnourished obese ESKD patients and malnourished obesity ESKD patients.

Author's reply: Thank you very much for your suggestion. We have included this issue as follows: “Overall, significant differences in baseline patient characteristics were observed among three groups (P < 0.05)”

comment 6: Paragraph 3.2 seems to concern different database:

“Out of the 1,244,415 hospitalized advanced CKD patients, 131,372 were diagnosed with malnutrition, resulting in an overall prevalence rate of 10.6%...”

Author's reply: Thank you very much for pointing this out. We apologize for the mistake and have removed the sentences from the revised manuscript.

comment 7: Whole manuscript needs careful edition, e.g. reference:

  1. Kalantar-Zadeh K, Rhee CM, Chou J, Ahmadi SF, Park J, Chen JL, et al. The Obesity Paradox in Kidney Disease: How to 425 Reconcile it with Obesity Management. Kidney Int Rep. 2017;2(2):271-81.

is repeated as ref, nr 17.

Author's reply: Thank you for highlighting this. We apologize for the error and have updated the references so that they are consistent (reference no. 15).

Reviewer 2 Report

Comments and Suggestions for Authors

Summary of the Study

This manuscript presents a large, nationally representative analysis of hospitalized end‑stage kidney disease (ESKD) patients using the U.S. National Inpatient Sample (NIS) from 2016–2021. The authors examine how obesity, with and without coexisting malnutrition, relates to in‑hospital mortality, adverse clinical outcomes, inpatient treatments, and healthcare resource utilization. The study addresses an important and understudied clinical question: whether malnutrition modifies the well‑described “obesity paradox” in ESKD.

The manuscript is well written, methodologically sound, and contributes novel insights. The use of a large national dataset strengthens the reliability of the findings, and the authors appropriately acknowledge the limitations inherent to administrative data.

However, several areas require clarification or refinement before the manuscript is suitable for publication.

Major Comments

  1. Please clarify how malnutrition severity categories were handled (mild, moderate, severe).
  2. There is some inconsistency in the reported sample numbers (e.g., 674,367 ESKD patients vs. 1,244,415 advanced CKD patients in later sections).
  3. Although the authors adjust for many covariates, several clinically relevant factors are not available (residual kidney function, dialysis adequacy, nutritional interventions, inflammatory markers). Please expand the discussion of residual confounding, especially regarding factors that may influence both malnutrition and mortality.
  4. The manuscript states that obesity is protective only in the absence of malnutrition. Consider clarifying that the observed “protective effect” is an association, not evidence of causality, given the cross‑sectional nature of the data.
  5. The manuscript repeatedly implies causal relationships despite using cross‑sectional administrative data, please address this in the manuscript

Minor Comments

  1. Methods Section

The authors use ANOVA for continuous variables but also report medians and IQRs for skewed data. Please clarify whether normality assumptions were met or whether non‑parametric alternatives were considered.

Author Response

Comment 1: Please clarify how malnutrition severity categories were handled (mild, moderate, severe).

Author's reply: Thank you very much for your insightful comment. We included all levels of malnutrition severity according to ICD-10 codes as follows;

Diagnosis

ICD 10 codes

Mild to moderate malnutrition

E44.0, E44.1, E46

Severe malnutrition

E40, E41, E42, E43, R64

These details have also been provided in Supplementary Table S1.

Comment 2: There is some inconsistency in the reported sample numbers (e.g., 674,367 ESKD patients vs. 1,244,415 advanced CKD patients in later sections).

Author's reply: Thank you very much for pointing this out. We apologize for the mistake and have removed the sentences from the revised manuscript.

Comment 3: Although the authors adjust for many covariates, several clinically relevant factors are not available (residual kidney function, dialysis adequacy, nutritional interventions, inflammatory markers).Please expand the discussion of residual confounding, especially regarding factors that may influence both malnutrition and mortality.

Author's reply: Thank you very much for your valuable comment and suggestion. We have addressed this point in detail in the discussion section of the revised manuscript.

“Although we adjusted for multiple potential confounders, several clinically relevant factors influencing the association between malnutrition and mortality in dialysis pa-tients were not captured in this nationwide database, including residual kidney function (RKF), dialysis modality, and nutritional interventions. RKF is a strong predictor of sur-vival, contributing to fluid and electrolyte balance and the clearance of middle molecules and protein-bound uremic toxins, with retention of these toxins increasing cardiovascular and infection-related mortality [22]. Patients with preserved RKF generally have better appetite and a less restricted diet, which may reduce the risk of malnutrition, whereas those without RKF often exhibit higher inflammatory markers [23], exacerbating the mal-nutrition–inflammation complex. Furthermore, dialysis modality such as conventional hemodialysis or hemodiafiltration, could not be determined; recent evidence suggests hemodiafiltration may improve survival compared with conventional high-flux HD [24]. Lastly, nutritional interventions, such as oral nutritional supplementation, can mitigate malnutrition and improve survival outcomes, particularly in hypoalbuminemic HD pa-tients [25]. Well-designed future studies incorporating these factors are warranted to better elucidate the relationship between malnutrition and mortality in dialysis patients.”

Comment 4: The manuscript states that obesity is protective only in the absence of malnutrition. Consider clarifying that the observed “protective effect” is an association, not evidence of causality, given the cross‑sectional nature of the data.

Author's reply: Thank you very much for pointing this out. We have revised the wording throughout the discussion and conclusion sections of the manuscript to emphasize association rather than imply a causal relationship.

Comment 5: The manuscript repeatedly implies causal relationships despite using cross‑sectional administrative data, please address this in the manuscript

Author's reply: Thank you very much for pointing this out. We have revised the wording throughout the discussion and conclusion sections of the manuscript to emphasize association rather than imply a causal relationship.

Comment 6: Methods Section

The authors use ANOVA for continuous variables but also report medians and IQRs for skewed data. Please clarify whether normality assumptions were met or whether non‑parametric alternatives were considered.

Author's reply: Thank you very much for pointing this out. We apologize for the incomplete clarification regarding the use of the Kruskal-Wallis test for skewed variables. This has been clarified in the revised manuscript. 

"Baseline characteristics, in-hospital outcomes, and resource utilization were compared among the obesity without malnutrition, obesity with malnutrition, and non-obese groups using one-way ANOVA for normally distributed continuous variables, the Kruskal-Wallis test for skewed continuous variables, and the Chi-squared test for categorical variables. "

Round 2

Reviewer 1 Report

Comments and Suggestions for Authors

It seems that the manuscript has been adequately reviewed and limitations highlighted.

Comments on the Quality of English Language

none

Author Response

It seems that the manuscript has been adequately reviewed and limitations highlighted.

response: Thank you

Reviewer 2 Report

Comments and Suggestions for Authors

All concerns have been addressed, thank you 

Author Response

All concerns have been addressed

Response: Thank you